# Genetic Factors Involved in Cardiomyopathies and in Cancer

**DOI:** 10.3390/jcm9061702

**Published:** 2020-06-02

**Authors:** María Sabater-Molina, Marina Navarro-Peñalver, Carmen Muñoz-Esparza, Ángel Esteban-Gil, Juan Jose Santos-Mateo, Juan R. Gimeno

**Affiliations:** 1Unidad de Cardiopatías Hereditarias, Servicio de Cardiología, Hospital Universitario Virgen dela Arrixaca, El Palmar, 30120 Murcia, Spain; mariasm@um.es (M.S.-M.); tanzi2001@hotmail.com (M.N.-P.); carmue83@gmail.com (C.M.-E.); jgimeno@secardiologia.es (J.R.G.); 2Universidad de Murcia, El Palmar, 30120 Murcia, Spain; 3Instituto Murciano de Investigación Biosanitaria (IMIB), El Palmar, 30120 Murcia, Spain; 4European Reference Networks (Guard-Heart), European Commission, 30120 Murcia, Spain; 5Red de investigación Cardiovascular (CIBERCV), Instituto de Salud Carlos III, 28029 Madrid, Spain; 6Biomedical Informatics & Bioinformatics Platform, Institute for Biomedical Research of Murcia (IMIB)/Foundation for Healthcare Training & Research of the Region of Murcia (FFIS), 30003 Murcia, Spain; angel.esteban@ffis.es

**Keywords:** cardiomyopathy, cardiotoxicity, arrithmogenic cardiomyopathy, dilated cardiomayopathy, cancer, genetic

## Abstract

Cancer therapy-induced cardiomyopathy (CCM) manifests as left ventricular (LV) dysfunction and heart failure (HF). It is associated withparticular pharmacological agents and it is typically dose dependent, but significant individual variability has been observed. History of prior cardiac disease, abuse of toxics, cardiac overload conditions, age, and genetic predisposing factors modulate the degree of the cardiac reserve and the response to the injury. Genetic/familial cardiomyopathies (CMY) are increasingly recognized in general populations with an estimated prevalence of 1:250. Association between cardiac and oncologic diseases regarding genetics involves not only the toxicity process, but pathogenicity. Genetic variants in germinal cells that cause CMY (LMNA, RAS/MAPK) can increase susceptibility for certain types of cancer. The study of mutations found in cancer cells (somatic) has revealed the implication of genes commonly associated with the development of CMY. In particular, desmosomal mutations have been related to increased undifferentiation and invasiveness of cancer. In this article, the authors review the knowledge on the relevance of environmental and genetic background in CCM and give insights into the shared genetic role in the pathogenicity of the cancer process and development of CMY.

## 1. Genetic Causes of Cardiomyopathies

Cardiomyopathies (CMY) are myocardial disorders in which the heart muscle is structurally and functionally abnormal in the absence of conditions that can explain the observed myocardial abnormality [1]. Mutations in sarcomeric, cytoskeletal, and desmosomal genes account for the vast majority of these monogenic conditions, which are inherited with an autosomal dominant pattern. CMY are characterized by genetic and clinical heterogeneity, with an estimated prevalence of 1:250 population and are associated with significant morbidity and mortality [2,3].

Hypertrophic cardiomyopathy (HCM) is the most prevalent CMY, which is followed by dilated (DCM), arrhythmogenic (ACM), and restrictive cardiomyopathy.

Cardiac phenotype in HCM commonly develops between the second to fourth decades of life, but it can flourish virtually at any age. The maximum increase in myocardial mass is reached typically within 2–3 years and then a plateau with mild decrease is observed over decades. Symptoms of dyspnoea and exercise limitation are associated withleft ventricular (LV) obstruction and diastolic dysfunction, while systolic function is preserved in most cases. Contribution of growth hormones, intense physical demand, and hypertension have been analyzed, but none of them have been definitively assumed as a significant trigger [4]. Apart from steroids that can facilitate the development of cardiac hypertrophy, other agents and toxicities do not seem to have a relevant impact in the natural history of HCM [5].

In contrast to HCM, in which the genetic nature of the disease was suspected from the initial descriptions [6], DCM has been thought to be secondary to a number of cardiac “injuries”. The process of myocardial wall thinning, LV dilatation, and systolic impairment has been traditionally understood as the consequence of viral infection, myocardial ischemia, increased cardiac volume conditions like valvular insufficiency, longstanding uncontrolled hypertension or arrhythmia, and toxicity from alcohol abuse or therapies [7,8,9,10,11,12,13,14,15,16,17] (Table 1). Nevertheless, the genetic factor was supposed to be an important cause of DCM, with a genetic test yielding around 40% with the current next-generation sequencing panels [18].

The development of high-throughput technologies for genetic testing has led to the identification of new genes associated with DCM as the recognition of titin (TTN) as one of the main DCM associated genes in 2012 [29,30]. Up to 25% of familial and 18% of sporadic DCM had pathogenic variants in TTN. Genetic variants in TTN and in other genes were soon identified in patients with secondary causes of DCM, alcohol abuse, peripartum, and cardiotoxic medications [7,8,9,10,11,12,13,14,15,16,17] (Table 1). The theory of the double or multiple genetic and environmental hits was accepted amongst scientific and clinicians [7,31] (Figure 1). Currently, the yield of the familial and genetic studies in DCM is similar to HCM. 

ACM is a disease characterized by the loss of myocardial tissue and subsequent fibro-fatty replacement [32]. Initial description of the disease supported the hypothesis of a right ventricle (RV) condition. LV involvement was then described as the disease progressed, and lately, typical primary LV forms have been recognized [32,33]. When LV function is preserved, the cardiac output is maintained and predominant symptoms are secondary to life threatening ventricular arrhythmia. Diagnosis of left forms of ACM are characterized by “atypical DCM” features such as early onset of arrhythmia despite mild-moderate degrees of systolic impairment and myocarditis-like pattern of late gadolinium enhancement (LGE) on cardiac magnetic resonance (CMR) (Figure 2). ACM is the most inflammatory idiopathic cardiomyopathy that can develop in the crisis of myocarditis [32,34].

Restrictive cardiomyopathy, compared to the former CMYs, is less likely to be a monogenic condition caused by mutations in the sarcomeric or cytoskeleton gene [35]. Primary or secondary amyloidosis accounts for the majority of this infiltrative disease of the myocardium [36]. The diffuse extracellular matrix increases in relation to the deposition of proteins like the wildtype of mutated transthyretin or circulating kappa or lambda light chains.

## 2. Cancer Therapy-Related Myocardial Dysfunction and Heart Failure

Advances in cancer treatment have led to the improved survival of patients with malignancies, but have also increased the related side effects [37,38]. Cardiovascular complications are one of the most frequent side effects. This may be the result of the direct effects of the cancer therapy on heart structure and function, or may be due to the accelerated development of cardiovascular disease. Chemotherapy drugs, alone or in association withradiotherapy, increase survival and lower the recurrence rate of cancer, but their use can be limited by cardiotoxicity. 

Cancer therapy-induced cardiomyopathy (CCM) can occur during, shortly after, or many years beyond cancer therapy, and may vary from subclinical myocardial dysfunction to irreversible heart failure (HF). Thus, in the long-term, the risk of death from cardiovascular causes exceeds that of tumor recurrence for some forms of cancer [39,40,41].

Cardiovascular complications of cancer treatment include myocardial dysfunction and HF, coronary artery disease, valvular disease, arrhythmias, arterial hypertension, thromboembolic disease, peripheral vascular disease, stroke, pulmonary hypertension, and pericardial complications [38,39,40,41,42]. Myocardial dysfunction and HF are nowadays the most commonly recognized cancer therapy-related cardiovascular complications. Table 2 summarizes the chemotherapy drugs associated with HF and myopericarditis [43].

### 2.1. Anthracyclines

Anthracyclines are a class of highly effective chemotherapy agents used for the treatment of many solid and hematologic cancers. The most commonly accepted pathophysiological mechanism of anthracycline-induced cardiotoxicity is the oxidative stress hypothesis, which suggests that the generation of reactive oxygen species and lipid peroxidation of the cell membrane damage cardiomyocytes [37,44]. 

The cardiotoxicity of anthracyclines may be acute, early, or late. Acute toxicity is rare (1%) and usually manifests as supraventricular arrhythmias, transient LV dysfunction, or electrocardiographic changes (QT prolongation). It develops immediately after infusion and is usually reversible. However, acute cardiac dysfunction may also reflect myocyte injury that can eventually evolve into early or late cardiotoxicity. There are no proven strategies to identify if cardiac dysfunction is reversible or progressive; however, the elevation of cardiac biomarkers may be a way to identify patients at risk for long-term cardiotoxicity [37,43]. Early effects (1.6–2.1%) occur within the first year of treatment, and late effects (1.6–5%)manifest after several years (median of seven years after therapy). Early and late toxicity are more likely to be irreversible, so early detection and treatment is of paramount importance [37,41,45,46].

Risk factors for anthracycline-related cardiotoxicity include lifetime cumulative dose. Doxorubicin is associated with a 5% incidence of congestive HF with a cumulative lifetime dose of 400 mg/m^2^ and 48% at a dose of 700 mg/m^2^ [47]. However, there is truly no safe dose of anthracyclines and HF rates can be up to 10% with standard doses in patients >65 years or with preexisting cardiovascular risk factors or cardiac diseases [43,48]. Factors increasing the risk of anthracycline toxicity include the presence of other cardiovascular disease risk factors, cardiac diseases associating increased wall stress, older age (>65 years) and pediatric population (<18 years), renal failure, associated therapies like mediastinal irradiation, and concomitant therapy with agents such as cyclophosphamide, paclitaxel, and trastuzumab [37,41,47,49].

In one study that included 2625 patients, the overall incidence ofcardiotoxicitywas 9% and in 98% of cases, cardiotoxicity occurred within the first year [50]. LV ejection fraction (EF) recovery and cardiac event reduction may be achieved when cardiac dysfunction is detected early and a modern HF treatment is promptly initiated. Conversely, if patients are identified late after the onset of cardiac dysfunction, HF is often difficult to treat [17]. 

### 2.2. Other Conventional Chemotherapies

Cyclophosphamide cardiotoxicity is relatively rare (generally occurring at higher doses >140 mg/kg) [51] and occurs within days of drug administration [52]. Other alkylating agents such as cisplatin and ifosfamide are uncommon causes of HF, usually due to volume overload during infusion (platin-containing chemotherapy requires the administration of a high intravenous volume to avoid toxicity). Volume overload in patients with pre-existing myocardial impairment is often the cause of HF, rather than the direct toxicity of these drugs.

Taxanes (docetaxel) interfere with the metabolism and excretion of anthracyclines and may potentiate left ventricular dysfunction risk, particularly with high dose anthracycline use, but the absolute cardiotoxic risks with taxanes are unknown [37,53]. Docetaxel also appears to increase HF risk in patients with preexisting cardiac diseases [54].

In several large-scale trials of adjuvant therapy in breast cancer, the rate of trastuzumab-related cardiac dysfunction ranged from 7 to 34%, with HF class III or IV rates between 0 and 4% [55]. The risk is higher in patients with preexisting cardiovascular diseases or hypertension and lower in anthracycline-free regimens [37]. In contrast to anthracyclines, trastuzumab cardiotoxicity typically manifests during treatment. Trastuzumab-induced LV dysfunction and HF are usually reversible with trastuzumab interruption and/or treatment with HF therapies [56]. The mechanism of anti-HER2 drug-induced cardiotoxicity includes structural and functional changes in contractile proteins and mitochondria, but it rarely leads to cell death, explaining the potential for reversibility [37,57].

Discussion on chemotherapy strategies should be taken very cautiously, particularly when a combination of potentially cardiotoxic drugs is needed. Future research is warranted for the development of effective preventive medication [58].

Inhibition of the vascular endothelial growth factor (VEGF) signaling pathway has been linked to hypertension, ischemia, LV dysfunction, and HF. The anti-VEGF antibody bevacizumab induced LV dysfunction in 2% of patients and HF (NYHA III or IV) in 1% of patients [59]. Similarly, cardiotoxicity was found for the TKIs (sunitinib, pazopanib, and axitinib) with cardiac dysfunction in 3–15% and symptomatic HF in 1–10% of patients [60,61].

### 2.3. Radiotherapy

Radiation induced heart disease may manifest years after exposure. The spectrum of pathology affecting the heart spans from acute to chronic and can affect almost all facets of the heart including, but not restricted to the pericardial sac, coronary arteries, myocardium, and heart valves [62]. LV dysfunction and HF can occur as acute radiation myocarditis, but more commonly develops as a long-term consequence of fibrosis, leading to ventricular dysfunction or restrictive cardiomyopathy [41,63]. The presence of other cardiovascular risk factors, concomitant anthracycline use, and anterior or left chest irradiation all increase risk. Mediastinal irradiation increases HF risk even 40 years after initial exposure [41,63].

## 3. Evaluation of Cardiac Injury from Chemotherapy

CCM or cancer therapy-induced cardiac dysfunction is defined as a decrease in the LVEF of 10%, to a value below the lower limit of normal [64]. 2D echocardiography is the method of choice for the detection of myocardial dysfunction. However, this technique has relatively moderate reproducibility, which can be improved by the use of 3D echocardiography [65]. At present, the value of deformation imaging for early detection of LV dysfunction is of great importance [66]. Thus, a reduction of global systolic longitudinal myocardial strain (GLS) accurately predicts a subsequent decrease in LVEF [67,68]. A relative percentage reduction of GLS of 15% from the baseline is considered abnormal and a marker of early LV subclinical dysfunction [38,67,68]. Other techniques such as CMR and nuclear cardiac imaging can also be used to evaluate the LVEF.

CMR is considered the gold standard for the measurement of ventricular volumes and function [69,70]. It also enables myocardial tissue characterization, providing useful information of the underlying histopathological changes, potentially allowing for the recognition of early myocardial injury [71]. Compared with echocardiography, CMR improves the detection of asymptomatic LV dysfunction in patients exposed to anthracycline chemotherapy and/or chest-directed radiation therapy (RT) [72]. Different studies have shownsubtle changes in the form of increased ventricular volumes or depressed systolic function, even before meeting the current criteria for cardiotoxicity [73], and as soon as one month after anthracycline initiation [19]. T1-based methods such as late gadolinium enhancement (LGE) or extracellular volume (ECV) quantification allow for the detection of myocardial fibrosis and scar, both during and at the end of therapy. Data on the prevalence, pattern, and prognostic significance of LGE in cases of anthracycline exposure are scarce and contradictory. In general, the prevalence of LGE in this setting seems to be low (<10% of cases), in contrast with ischemic cardiomyopathy (up to 100%) or in patients with idiopathic DCM (≈45%) [74]. The patterns of LGE described in anthracycline treated patients include subepicardial, midmyocardial, and RV insertion point. Combined anthracycline and trastuzumab therapy has led to a greater frequency of LGE in studies than the use of anthracycline alone, and it is predominantly associated with a subepicardial pattern [24,25,26,75] (Table 1). The low sensitivity of LGE, a marker of focal fibrosis, is explained by the diffuse nature of interstitial fibrosis in this setting, which has been confirmed from pathology exams [76]. Diffuse fibrosis can be detected by a relatively novel CMR modality of extracellular volume fraction (ECV) quantification. ECV has been recently correlated with anthracycline dose, functional capacity, LV dysfunction, and markers of adverse ventricular remodeling in pediatric [77] and in adult patients [78] after completion of anthracycline therapy. Other novel techniques like T2 mapping seem to be promising in this field. Recent data based on experimental models show T2 mapping as the earliest marker of anthracycline-induced cardiotoxicity in the absence of T1 mapping, ECV, or LV motion abnormalities [79].

## 4. Impact of Chemotherapy on Development of Systolic Impairment in Patients with Germinal Mutations in Cardiac Associated Gene Loci

Traditionally, DCM was characterized by an enlarged and poorly contractile LV. The degree of LV systolic dysfunction can be variable and the disease is often progressive. Hypokinetic non-dilated and dilated normocontractile LV forms of DCM have been recently included as a part of the spectrum of DCM [80]. Hypokinetic non-dilated can be the mode of presentation in most acute and early forms of cardiotoxicity.

DCM can be attributed to genetic and non-genetic causes including hypertension, valve disease, inflammatory/infectious causes, and toxins [13,81,82]. Even these “nongenetic” forms of CMY can be influenced by an individual’s genetic profile, and furthermore, mixed etiologies may be present [1,80,82].

DCM is genetically heterogeneous. Likely pathogenic genetic variants in up to 50 different genes have been associated with DCM, and this number continues to grow. The involved genes encode cytoskeletal, sarcomeric, mitochondrial, desmosomal, nuclear membrane, and RNA binding proteins [1,82]. The gene that encodes titin—the giant protein that controls the stiffness of the sarcomere—is the most common and is responsible for ≈20% of cases of familial DCM [1,29]. Titin truncating variants, via titin haploinsufficiency, are the main cause of familial DCM [83].

The prevalence of familial DCM, in the global DCM cohorts, is assumed to be around 30–50% (Table 1) [13,81,82,84,85].In patients with familial DCM, approximately 40% has an identifiable genetic cause [82,85].The mode of inheritance is usually autosomal dominant with variable expressivity and penetrance, but autosomal recessive, x-linked, and mitochondrial inheritance have also been described. De novo mutations also contribute to genetic cardiomyopathy and are defined when neither biological parent carries the offspring’s mutation.

The left form of ACM can also present like DCM (in early phases as hypokinetic non-dilated form). The arrhythmic burden in ACM compared to DCM is higher. Accurate diagnosis of a prior ACM is important not only because of the likely impact of the myocardial toxicity of chemotherapy, but also because of the life threatening arrhythmias that can precede the development of severe systolic impairment [32]. A high suspicion of ACM should be taken when there is a CMY, history of ventricular arrhythmia, and family history of sudden death. Of note, the pattern of LGE on CMR is typically very similar to that of myocarditis and cardiotoxicity from chemotherapy (Table 1 and Figure 2). In the absence of a baseline CMR, evaluation of cardiotoxicity after chemotherapy can be difficult. Patients with a personal or familiar diagnosis of CMY undergoing chemotherapy should have a full cardiac evaluation including a baseline CMR. An illustrative case of a young woman with CCM in whom following familial evaluation was subsequently diagnosed with ACM caused by a DSP p.Q447* is presented in Figure 3.

Elucidating whether an underlying genetic condition is present in the evaluation of CCM can be challenging. There is little information on the systematic evaluation of patients with CCM in which genetic or familial study have been performed [27,86,87,88,89]. Despite genetic testing beingconsidered as a routine diagnostic and prognostic exam in oncologic patients, genetic testing of CMY associated genes (germinal cells) is not included in the panels.

TTN-truncating variants have been associated with the development of CCM in patients with chemotherapy [27]. TTN seem to be the most prevalent in all causes of DCM, from familial, peripartum, alcoholic, and also CCM. A recent publication reported the results from the analysis of 213 patients (99 diverse cancers, 73 breast cancer, 41 children with acute myeloid leukemia) who were mainly treated with anthracyclines (90% and 33% of adults received trastuzumab) undergoing genetic study and found the prevalence of 7.5% of patients with a titin-truncating variant.

Additionally, rare variants in genes related to familial CMY have been identified in sporadic individuals and a small case series of patients with CCM [27,86,87,88,89]. The current evidence is based on clinical cases like a woman with epirubicin-induced CMY who was found to be the carrier of the mutation in a sarcomeric gene (MYH7) often associated with HCM or DCM phenotypes [87]. MYH7 variants were also identified by other authors in patients developing severe CCM [86,89]. Some of the common denominators in the clinical features of those cases were an unusual severity of the systolic impairment, unexpectedly soon after therapy and young age. CMR features in patients with genetic variants in HCM associated genes can be a particular pattern of LGE on CMR (Figure 2). A familial form of CCM was suggested in the description of the families in an interesting publication from Wasielewski et al. [89].

Genome wide association (GWAS) studies have identified a number of loci with enriched variants in patients with CCM compared to the controls [90,91]. Candidate markers were not located in known CMY related genes. Further investigation including functional studies is warranted in order to identify implicated genes. Future studies might shed light not only on the pathogenicity of CCM, but also on the identification of new CMY associated genes. Interestingly, another recent publication has demonstrated the significant association of a number of rare and common polymorphisms in 72 CMY genes in patients with CCM [92]. Within the list of genes, OBSCN seemed to harbor a higher number of missense variants in coding fragments in CCM.

## 5. Predisposition to Cancer in Patients with Cardiac Associated Genetic Variants

### 5.1. Arrhythmogenic Cardiomyopathy: Disease of the Desmosome

ACM is a major cause of sudden death in European countries, particularly among individuals younger than 35 years and in athletes [93]. Although initially described as a disease affecting the RV, this condition can affect both ventricles or produce a predominant involvement of the LV [94].

This disease is caused by a loss of integrity of desmosomes, organelles fundamental not only in maintaining the integrity of the junctions, but also in the regulation of metabolic pathways and in the maintenance of gap junctions. While genetic studies have demonstrated the presence of a causal mutation in 40–50% of patients [95], immunohistochemical studies performed in patients with arrhythmogenic RV cardiomyopathy showed an alteration in the expression of plakoglobin (protein that is part of the intercellular desmosome bond) in 80% of cases [96].

Plakoglobin migration from the junctions to the cell nucleus activates signaling pathways involved in the degeneration of cardiac muscle with fibrofatty tissue [97]. While the fibro-fatty replacement is the substrate for the initiation and maintenance of ventricular arrhythmias, typical arrhythmogenic, remodeling of GAP junctions (particularly altered expression of connexin 43) also favors the occurrence of arrhythmias [98]. Inflammation is another essential element in the pathophysiology of ACM. It has been recently shown that an increase in cytokine production in the myocardium of patients with ACM as well as the elevation of certain serum cytokines [99].

### 5.2. Inherited Cardiac Disease and Cancer

The risk of hematologic or solid cancer is higher in the so called “Rasopathies”. Noonan syndrome, Costello, and other similar syndromes characterized by specific dysmorphic features and cardiac involvement with HCM development during infancy are caused by genetic mutations in a set of genes of the RAS-MAPK group (PTPN11, KRAS, SOS1, RAF1, SHP2) [100] (Figure 4).

On the other hand, mutations in the gene for lamin A/C (LMNA), a nuclear membrane protein known to cause skeletal disorders, progeria, muscular dystrophy, and DCM, are also associated with an increased risk for the development of different types of cancer. Interestingly, cardiac histology in patients with DCM mutations in the LMNA gene is similar to that of ACM caused by the effect of a mutation in a desmosomal gene [101]. Similar to desmosomal ACM, in laminopathies, a high risk of ventricular arrhythmias and sudden death have been reported.

Table 3 shows a list of the main genes related to CMY and the frequency of total somatic mutations and in the different tumor tissues obtained from the COSMIC database [102]. We can observe a high rate of somatic mutations in many of these genes, which may suggest a putative effect of these mutations on the carcinogenesis. The genes harboring the highest frequency of somatic mutations are TTN, DMD, and DSG2. The TTN and DMD associated CMY phenotype is DCM, while DSG2 is ACM. Both TTN and DMD are two of the largest three genes in humans, but DSG2 is a small desmosomal gene (5652 bp of the transcript).

A dedicated search of the interaction between selected CMY associated genes and molecular pathways implicated in cancer based on information from the KEGG (Kyoto Encyclopedia of Genes and Genomes terms) is represented in a chord chart in Figure 5. From the total of 33 genes initially included (listed in Table 3), 25 links were identified with 17 metabolic pathways (Appendix A). Apart from PTPN11 and LMNA, another 12 genes from sarcomeric (thin and thick filament), desmosomal (PKG/JUP), metabolic (PRKAG2, LAMP2, GLA), and calcium handling (PLN) as well asothers showed interactions. Interestingly, encoded proteins from this list of genes were linked to relevant cellular pathways from signaling, metabolic, adhesion, apoptosis, and toxicity. Further studies are needed to elucidate the role of genetic variants in these and other candidate genes.

To visualize the relationships between the cardiac genes and the cancer pathways, we used a graph using the R library described in [110]. In this case, instead of representing gene ontology (GO) terms, we represented the KEGG pathways.

### 5.3. Desmosome and Its Association with Invasiveness of Cancer Cells

The complex formed by the desmosome and intermediate filament comprises cadherins (desmogleins: encoded by DSG1-4 and desmocollines: DSC1-3), placophilines (PKP1-4) plakoglobin (PKG or γ- catenin), and desmoplakin (DSP2) [111]. Desmoplakin (DSP) is an obligate desmosomal plaque component [112,113] andinteracts with the DSP, plakoglobin (γ-catenin), plakophilin, and desmin intermediate filaments, providing an intimate bond between the desmosomal cadherins and cytoskeleton [114,115].

The role of desmosomal components in cancer progression is being revealed, but it is still largely unknown [111]. Several studies have already suggested that the reduction in the number of desmosomes may influence epithelial cell invasion and metastasis [116,117], since an important function of desmosomes related to cancer is their ability to inhibit cell motility [118]. A reduced expression of desmosomal proteins in breast cancer [119], in oropharyngeal squamous cell carcinoma [120], uterine cervix cancer [121], colorectal cancer [122], and in pancreatic cancer has been reported [123].

Recently, studies have shown abnormal expression of desmosomal proteins in different types of cancers. For example, the association of primary colorectal cancer and low DSC3 activityhas been reported, which is associated with worse prognosis. P53 transfection of tumor explants increased the expression of DSC3 in some studies [122]. In breast cancer, a decrease in the levels of DSC3 has been observed [119]. The aberrant expression or disruption of DSC2 might lead to heart disorders, certain cancers, and some other human diseases [124]. A reduction in the expression of a desmosomal protein DSC2 has been associated with shortened survival, high-grade malignancy and lymph node positivity in pancreatic tumor [123], and a decrease in DSC2 in colorectal cancer has also been observed [125]. DSG3 is overexpressed in squamous cell carcinoma and head and neck cancer, and abnormal expression of DSG2 in melanoma, squamous cell carcinoma, and gastric cancer has also been reported [121]. A putative role of DSG2 as a tumor suppressor in human breast cancer has been suggested [126], and its expression is related to the tumor size, lymph node metastasis, and stage in lung adenocarcinoma [127].

Furthermore, plakophilin expression is altered in various cancer types (lung and prostate cancer) and can be correlated with the patients’ survival [128]. Oropharyngeal tumor samples had a reduction in the expression of desmoplakin [120].

Some studies have shown an early role of DSP function reduction in carcinogenesis [116,129]. DSP acts as a tumor suppressor by inhibiting the Wnt/β-catenin signaling in non-small cell lung cancer [130]. DSP expression was downregulated in eightout of 11 (73%) cell lines and in 34 of 56 (61%) primary lung tumors [130]. DSP overexpression facilitates plakoglobin (γ-catenin) expression, resulting in a reduction in T-dependent transcriptional activity/lymphoid facilitating (TCF/LEF) cell factor and a reduction in the expression of target genes of Wnt/β -catenin as AXIN2 and MMP14 matrix metalloproteinase. In contrast, deletion of DSP by miRNA interference resulted in downregulation of plakoglobin and upregulation of β-catenin and MMP14. These data suggest that DSP is inactivated in lung cancer by an epigenetic mechanism, leading to an increased sensitivity to apoptosis by cancer therapy, acting as a suppressor of the function of tumor, possibly through inhibition of the signaling pathway Wnt/β-catenin in non-small cell lung tumors. Epigenetic regulation of DSP and its ability to increase the sensitivity to apoptosis of cancer therapy have important implications for clinical application [130].

The Wnt signaling pathway has been mentioned several times in relation to different types of cancer, since it controls proliferation and differentiation processes, both crucial when focusing on cancer. In the last years, a body of evidence has supported the role of the Wnt pathway in the development of fibro-adipose myocardial substitution in patients with desmosomal mutations causing ACM [97,131,132]. Among the catenins, β-catenin, which activates the Wnt-signaling pathway [133], is involved in cellular adhesion, growth, and differentiation and has been implicated in the transition of normal cells to transformed/cancer cells [134], highlighting a tumor suppressor role of β-catenin, similar to other adherens junction proteins, in maintaining junctional integrity [135].

Up to 10% of patients with pancreatic ductal carcinoma presented a family history of the disease. Germinal mutations in BRCA2, p16/CDKN2A, STK11, and PRSS1 have been associated with an increased risk of pancreatic ductal carcinoma [136]. Pancreatic ductal carcinoma is the tumor with the highest incidence of somatic mutations in KRAS, which also presents somatic variants in the Wnt pathway-β-catenin [137]. The role of DSP mutations both germ and somatic in the development of pancreatic ductal carcinoma has not been studied. The deletion of DSP induced tumor microinvasion in genetically modified mice, developing neuroendocrine pancreas cancer [116]. One study identified a germline missense variant in DSG2 in a familial gastric cancer patient and no somatic mutations were identified [138].

Despite the attractive hypothesis of the role of desmosomal mutations in the pathogenicity of cancer, it is still weak. Studies summarized earlier have shown striking increase or decrease in desmosomal components in different types of cancer, but still, different authors have presented contradictory results. Within the desmosomal plakoglobin, (PKG/JUP) and DSG2 seem to be the most likely candidate genes to study their pathogenicity in cancer development. Plakoglobin has been the only desmosomal gene involved in pathways in cancer in the dedicated search presented in Figure 5. Further studies involving animal models are needed to improve the understanding of the pathophysiology and the role of mutations in desmosomal genes in oncology [111]. The question of whether mutations in other CMY associated genes, apart from RAS/MAPK and LMNA [139], can impact on vulnerability of cancer remains unanswered.

## 6. Importance of Genetic and Family Study in Patients with Severe Cardiotoxicity

Family study is an important diagnostic part in the evaluation of patients with CMY. A familial disease can be demonstrated in 20% to 35% of cases when first-degree relatives are screened with an electrocardiogram and echocardiogram [140,141,142]. The percentage of familiar DCM can be even greater than initially expected, as high as 48%, when LV enlargement is considered as an early sign of the disease [80,143,144]. Inclusion of genetic testing helps to identify relatives at risk, for whom preventive interventions to slow progression and reduce complications can be recommended [13,80,144]. Other benefits from family screening are related to the definition of the disease in the proband and its causes, assessing the role of other genetic and environmental factors [13,80,145]. Valuable information on the disease expression is often taken from a comprehensive study of relatives including elder generations who have had more chances to be in contact with environmental triggers [145,146,147]. The American College of Medical Genetics and Genomics (ACMG) recommend a detailed 3-generational family history for the clinical practice of cardiomyopathy as well as the referral of patients to expert centers as needed, genetic counseling of patients and families by genetic counselors, and physicians with expertise in genetic cardiomyopathies and therapy based upon phenotype including drugs, devices, and special clinical recommendations by gene [148].

Recent studies using massive sequencing has shown that up to 50% of non-ischemic CMY is genetically determined [80,149]. More than 50 genes encoding for sarcomeric proteins, cytoskeleton, nuclear envelope, sarcolemma, ion channels, and intercellular junctions have been implicated in DCM and ACM. A genetic diagnosis allows for cascade testing of at-risk relatives and consideration of reproductive testing options [150]. 

As has been presented earlier in this paper, genetic predisposition and family history of CMY are potential risk factors to develop CCM [89]. The presence of oncological or cardiovascular factors listed in Figure 6 when assessing a patient with CCM should prompt further cardiac evaluation. The occurrence of CCM in the absence of a typical cardiotoxic agent, or if it is a low dose, particularly when it is not associated with radiotherapy of the chest and has no other comorbidities, then a high suspicion of other non-chemotherapy causes should be explored. Ischemic heart disease should be ruled out as the main cause of systolic impairment and HF in older patients with cardiovascular morbidities.

Amongst the CCM factors, an underlying CMY should be considered when there is significant systolic impairment (LVEF < 45%), ECG findings suggestive of an underlying CMY, atypical course of the disease with incomplete response to therapy, high arrhythmic burden, peculiar patterns of LGE or extensive fibrosis, or features of myopathy or rasopathy [89].

In conclusion, from the cardiogenetic perspective, based on the high prevalence of inherited CMY in the general population and that genetic testing is part of the routine work-up of patients with different types of cancer, a careful look at CMY associated genes should be promoted. Genetic testing may change the management of patients with cancer preventing CCM. Specific recommendations includedin future guidelines would be highly appreciated [37,80,151,152,153,154].

## Figures and Tables

**Figure 1 jcm-09-01702-f001:**
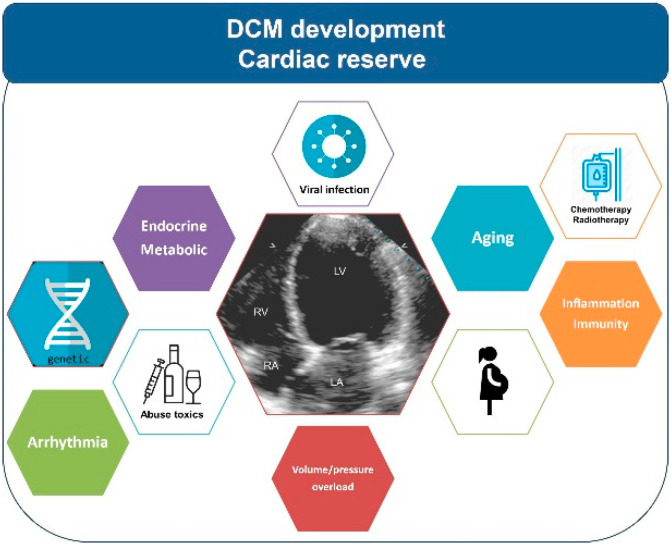
Schematic representation of the interplay of environmental and genetic factors participating in the development of dilated cardiomyopathy (DCM).RV: right ventricle; LV: leftventricle; RA: right atrial; LA: left atrial.

**Figure 2 jcm-09-01702-f002:**
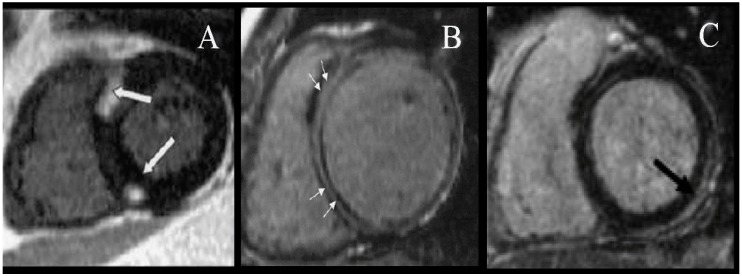
Late gadolinium enhancement patters seen in cardiomyopathies. (**A**) Typical left and right ventricular junction pattern, characteristic of hypertrophic cardiomyopathy (HCM), (**B**) midmyocardial lineal pattern characteristic of idiopathic/genetic dilated cardiomyopathy (DCM), (**C**) subendocardial left ventricle pattern characteristic of myocarditis and left form of arrhythmogenic cardiomyopathy (ACM), which can also be seen in low proportion of cases with cancer therapy-induced cardiomyopathy (CCM).

**Figure 3 jcm-09-01702-f003:**
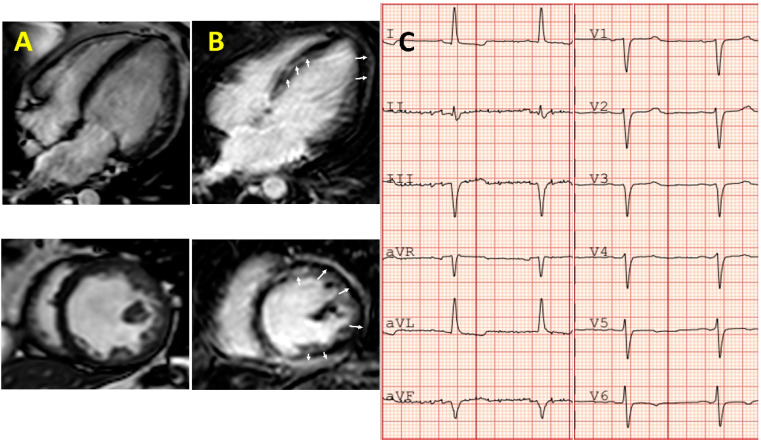
A 56-year-old female with cancer therapy-induced cardiomyopathy (CCM) (primary breast large B-cell lymphoma 10 years before treated with cyclophosphamide, hydroxy-daunorubicin, vincristine sulfate, prednisone, and radiotherapy). After familial evaluation was diagnosed with arrhythmogenic cardiomyopathy (ACM) caused by a DSP p.Q447*. (**A**) Cardiac magnetic resonance: four-chamber (up) and short-axis (down) views showing a mildly dilated left ventricle (LV)(end-diastolic volume 185 mL, 104 mL/m^2^), with LV hypertrabeculation from base to apex and severely impaired systolic function (left ventricle ejection fraction 29%). (**B**) LV global subepicardial late gadolinium enhancement (it is seen like a white ring). (**C**) Electrocardiogram showing a rS-type complex from V1 to V6, mild ST segment depression and negative T waves in V6, I, and aVL.

**Figure 4 jcm-09-01702-f004:**
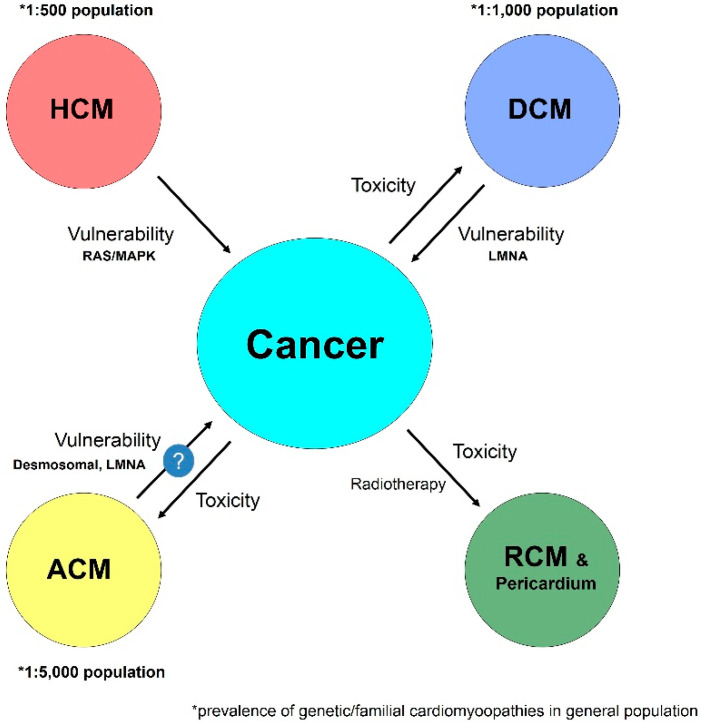
Schematic representation of the interaction between different cardiomyopathies (CMY) and cancer. HCM:Hypertrophic cardiomyopathy; ACM: Arrhythmogenic cardiomyopathy; DCM: dilated cardiomyopathy; RCM: Restrictive cardiomyopathy; LMNA: lamin A/C mutation; RAS/MAPK: RAS/mitogen activated protein kinasevariants.

**Figure 5 jcm-09-01702-f005:**
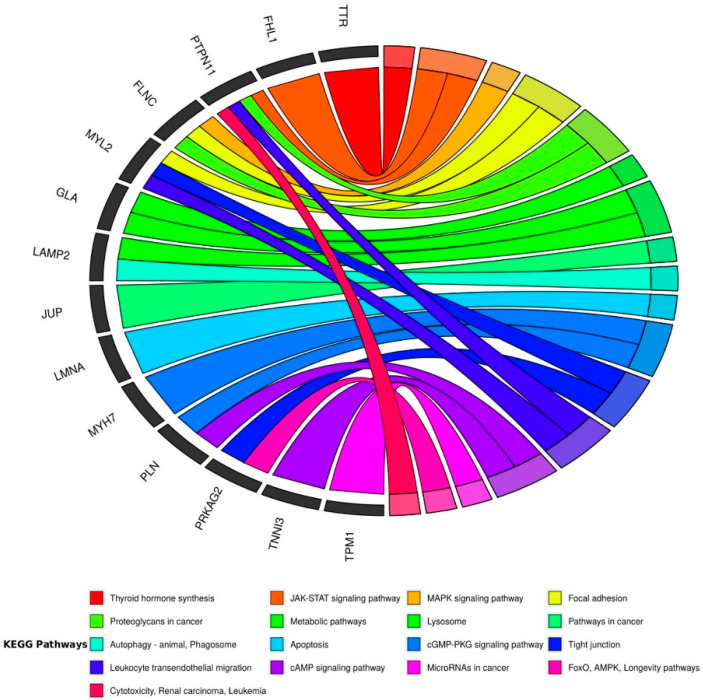
Chord diagram clustered by colors. Colors in the chords correspond to KEGG (Kyoto Encyclopedia of Genes and Genomes terms) pathways, containing the genes involved with the main cardiomyopathies (33 genes listed in Table 3), and were crossed with the KEGG database [108]. The results of 57 KEGG pathways were analyzed, filtered by oncology/cancer/tumor terms, and the connections were plotted. KEGG database covers information at different molecular levels. The KEGG Pathways database is a collection of manually curated pathways including information on molecular interactions, reactions, and network relationships. The KEGG database contains more than 24 million annotated genes and 530 pathways with more than six million pathway-linked genes [109].

**Figure 6 jcm-09-01702-f006:**
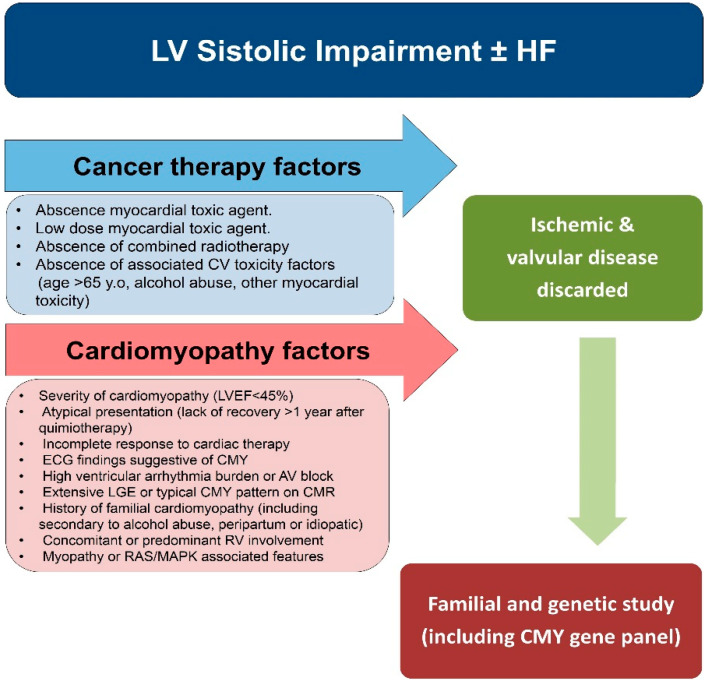
Proposal of a diagnostic algorithm in the evaluation of Cancer therapy-induced cardiomyopathy (CCM) with suspicious inherited cardiomyopathy. LV: left ventricle; HF: heart failure; CV: cardiovascular; CMY: Cardiomyopathies; LGE: late gadolinium enhancement, ECG: Electrocardiogram; CMR: cardiac magnetic resonance; LVEF: left ventricle ejection fraction; RAS/MAPK: RAS/mitogen activated protein kinasepathway; RV: right ventricle; AV: atrioventricular.

**Table 1 jcm-09-01702-t001:** Epidemiological, clinical, cardiac magnetic resonance (CMR), and genetic characteristics of different cardiomyopathies.

Type of Cardiomyopathy	Prevalence	DCM (%)	Family History of CMY (%)	LVEF Recovery (%)	LGE Prevalence (%)	Most Common LGE Distribution/LGE Pattern*	Other CMR Features	Death or HTx (%)	Gene	Genetic Variants (%)
Idiopathic Dilated Cardiomyopathy [7,19,20,21]	1:250–1:1000	30–50	30–50		30–65	Interventricular septum/Linear,	- ↑LVEDV, ↑LVESV, ↓LVEF - Diffuse LV wall thinning -Shortened postcontrast T1		BAG3 DES DMD FLNC LMNAMYBPC3 MYH7 PLNRBM20SCN5A TNNT2 TTN	40–50
					35 TTN mutation	Mid-wallInterventricular septum/Linear				
					88 LMNA mutation	Mid-wallBasal or mid-ventricular septal wall/Predominantly Epicardial				
					NA PLN mutation	Posterolateral LV wall/Predominantly epicardial				
Alcoholic Cardiomyopathy [8,9,11]		21–47	42	37–50	8			21–33	BAG3 LMNA MYH7 TTN	13.5 (9.9TTN)
Peripartum Cardiomyopathy [12,13,14]	1:100–1:4000	6	10–15	45–78				7–21	MYBPC3 MYH7 SCN5A TNNT2 TTN	15–20 (9.9 TTN)
Tachycardia-Induced Cardiomyopathy [13,15,16]				10–82				8–12	SCN5A	
Myocarditis [13,22,23]					44–93	LV inferolateral wall (PBV19)- anteroseptal(HHV6)/Subepicardial patchy or mid-wall	Myocardial edema in T2-STIR, EGE, pericardial effusion			
Cancer therapy-related myocardial dysfunction [17,24,25,26,27,28]				42–68	<10 (higher if trastuzumab adjuvancy)	Inferolateral wall/Subepicardial	- Early injury: ↑LVESV, ↓LVEF, ↓ LVMI, EGE +, T1 mapping +/-, ECV +/-, T2 mapping +		MYH7 TTN	7.5
							- During/early post-therapy:↑LVESV, ↓LVEF, LGE +			
							- Late cardiotoxicity: ↑LVESV, LVEF, LGE +, ECV+			

CMP: cardiomyopathy. CMR: Cardiac magnetic resonance. DCM: Dilated cardiomyopathy. HTx: Heart transplant. LGE: late gadolinium enhancement. LVEF: left ventricle ejection fraction. LVEDV: Left ventricle end-diastolic volume. LVESV: Left ventricle end-systolic volume. LVMI: Left ventricle mass index. EGE: Early gadolinium enhancement. ECV:Extracellular volume. STIR: Short Tau inversion recovery. % DCM indicates percentage of DCM caused by each specific etiology. NA: Not available.

**Table 2 jcm-09-01702-t002:** Cardiovascular toxicity associated with chemotherapy.

Cardiovascular Toxicity	Chemotherapy Agents
Heart failure	• Anthracyclines *(dose dependent):* doxorubicin, daunorubicin, idarubicin, epirubicin, mitoxantrone
	• Alkylating agents: cyclophosphamide, ifosfamide
	• Antimicrotubule agents: docetaxel
	• Monoclonal antibodies: trastuzumab, bevacizumab
	• Small molecule tyrosine kinase inhibitors: sunitinib, pazopanib, sorafenib, imatinib, dasatinib, lapatinib, nilotinib
	• Proteasome inhibitors: carfilzomib, Bortezomib
Myopericarditis	• Alkylating agents: Cyclophosphamide
	• Antimetabolites: 5-fuorouracil, cytarabine
	• Monoclonal antibodies: Trastuzumab, rituximab
	• Cytokines: Interleukin-2
	• Immune-checkpoint inhibitors

**Table 3 jcm-09-01702-t003:** Main genes involved in cardiomyopathies. Associated cardiac phenotype and prevalence of variants in different types of cancer.

Gene	Protein	Locus	Main Cardiac Disease	Frequency (%)	Ref.	Somatic Mutations in Cancer (%)	Main Tissues	% of Mutated Samples per Specific Tissue *	Comments	Ref. OMIM
*ACTC1*	*Actin α-cardiacmuscle 1*	*NG_007553*	*HCM, DCM*	1	[103]	1.08	Skin	6.47		* 102540
*BAG3*	*BAG cochaperone 3*	*NG_016125*	*DCM*	-		0.91				* 603883
*DES*	*Desmin*	*NG_008043*	*HCM, DCM, ACM*	<1	[82,104,105]	0.91	Skin	3.07		* 125660
*DMD*	*Dystrophin*	*NG_012232*	*DCM*			13.09	Cervix	10.5		* 300377
							Endometrium	10.84		
							Large intestine	6.89		
							Lung	6.56		
							Parathyroid	6.9		
							Skin	11		
							Soft tissue	6.62		
							Stomach	7.83		
							Urinary tract	7.73		
*DSC2*	*Desmocollin 2*	*NG_008208*	*ACM*	1–8	[104,105]	1.57				* 125645
*DSG2*	*Desmoglein 2*	*NG_007072*	*ACM*	3–20	[104,105]	17.16	Skin	4.37		* 125671
*DSP*	*Desmoplakin*	*NG_008803*	*ACM, DCM*	1–15, 2	[82,104,105]	3.55	Cervix Skin	5 10.03		* 125647
*EMD*	*Emerin*	*NG_008677*	*DCM*	-		0.34				* 300384
*FHL1*	*Four and a half LIM domains 1*	*NG_015895*	*HCM*	-		1.30				* 300163
*FHOD3*	*Formin homology 2 domain containing 3*	*NG_042837*	*HCM*	-		6.14	Skin	3.56		* 609691
*FLNC*	*Filamin C*	*NG_011807*	*ACM, DCM*	1	[82,103]	4.20	Cervix Peritoneum Skin	5.5 10.53 12.14		* 102565
*GLA*	*Galactosidase alpha*	NG_007119	*HCM*			0.55			Fabry disease	* 300644
*JUP*	*Junction plakoglobin*	*NG_009090*	*ACM*	0–1	[104,105]	1.39	Stomach	7.63 (CNV)	Naxos disease	* 173325
*LAMP2*	*Lysosomal associated membrane protein 2*	*NG_007995*	*HCM*	0.7–2.7	[106]	1.06			Danondisease	* 309060
*LMNA*	*Lamin A/C*	*NG_008692*	*DCM, ACM*	4–8, 3–4	[82,104,105]	1.16	Thyroid Breast	3.18 3.59 (CNV)	Charcot-Marie-Tooth disease, Emery-Dreifuss muscular dystrophy, Hutchinson-Gilford progeria, Malouf syndrome	* 150330
*MYBPC3*	*Myosin binding protein C cardiac*	NG_007667	*HCM, DCM*	35–40	[82,103]	1.75	Thymus	3.57		* 600958
*MYH7*	*Myosin heavy chain 7*	NG_007884	*HCM, DCM*	40–44, 3–4	[82,103]	3.67	Skin	11.09		* 160760
*MYL2*	*Myosin light chain 2*	NG_007554	*HCM*	1–2	[103]	0.71				* 160781
*MYL3*	*Myosin light chain 3*	NG_007555	*HCM*	1	[103]	0.31				* 160790
*PKP2*	*Plakophilin 2*	*NG_009000*	*ACM, DCM*	20–45,	[104,105]	2.54	Skin	4.69		* 602861
*PLN*	*Phospholamban*	NG_009082	*ACM, DCM*	0–12	[104,105]	0.33				* 172405
*PRKAG2*	*AMP-activated protein kinase subunit-γ-2*	NG_007486	*HCM*	1	[103]	3.96	Skin Stomach	3.01 (CNV) 3.05 (CNV)	Wolff-Parkinson-White syndrome, Glycogen storage disease of heart	* 602743
*PTPN11*	*Protein tyrosine phosphatase non-receptor type 11*	NG_007459	*HCM*	-		2.09	Penis	4.76	LEOPARD syndrome, Leukemia, juvenile myelomonocytic somatic, Noonan syndrome	* 176876
*RBM20*	*RNA binding motif protein 20*	*NG_021177*	*DCM*	2	[82]	2.70	CNS Skin	3.89 (CNV) 3.88		* 613171
*SCN5A*	*Sodium voltage-gated channel alpha subunit 5*	*NG_008934*	*DCM*	2–3	[82]	4.78	Skin	15.37	Brugada syndrome, Long QT syndrome	* 600163
*TMEM43*	*Transmembrane protein 43*	*NG_008975*	*ACM*	<2	[104,105]	0.60	Urinary tract	3.02 (CNV)	Emery-Dreifuss muscular dystrophy	* 612048
*TNNC1*	*Troponin C1. slow skeletal and cardiac type*	NG_008963	*HCM, DCM*	0.61, 0.24	[107]	0.26				* 191040
*TNNI3*	*Troponin I3 cardiac type*	NG_007866	*HCM, DCM*	1.35–5, 0.57	[103,107]	0.57				* 191044
*TNNT2*	*Troponin T2 cardiac type*	*NG_007556*	*HCM, DCM*	5–15, 3	[82,103]	0.89	Breast Thyroid	6.65 (CNV) 5.26 (CNV)		* 191045
*TPM1*	*Tropomyosin 1*	NG_007557	*HCM, DCM*	3, 1–2	[82,103]	0.94				* 191010
*TRIM63*	*Tripartite motif containing 63*	*NG_033268*	*HCM*	-		0.72				* 606131
*TTN*	*Titin*	*NG_011618*	*DCM*	12–25	[82]	24.49	Biliary tract	14.26		* 188840
							Breast	14.34		
							CNS	9.74		
							Cervix	26.5		
							Endometrium	25.49		
							GT	100		
							Large intestine	26.49		
							Liver	19.67		
							Lung	25.55		
							Meninges	7.02		
							Esophagus	16.92		
							Ovary	18.1		
							Pancreas	9.6		
							parathyroid	8.33		
							Peritoneum	5.56		
							Pleura	6.94		
							Prostate	8.5		
							Salivary gland	5.26		
							Skin	42.31		
							Stomach	33.64		
							Testis	5.62		
							Thyroid	10.54		
							UAT	20.76		
							Urinary tract	36.14		
*TTR*	*Transthyretin*	NG_009490	*HCM*	-		0.37			Amyloidosis	* 176300

Homo sapiens: GRCh38.p13 (GCF_000001405.33); * Filtered by confirmed somatic pathogenic mutation, in tumor sample >5% or the highest value when this was <5%. Shaded data were obtained from COSMIC database. Ref: References; CNV: Copy Number Variation; CNS: Central nervous system; UAT: upper aerodigestive tract; GT: gastrointestinal tract.

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
