# Peer review of "Genetic Factors Involved in Cardiomyopathies and in Cancer"

_jcm, 2020, doi:10.3390/jcm9061702_

Round 1

Reviewer 1 Report

I thank the authors for making the effort to review this interesting and valid topic, particularly to give background on cardiomyopathy genetics and cancer genetics/pathways in the same review. I would suggest the following changes:

Lines 66-81, section on DCM. Points to be addressed: 

  1. It is true that DCM was previously (in the past) thought to be secondary to causes other than genetic, but there were several papers suggesting this was not the case even before next generation sequencing technologies were available.
  2. TTN was not shown to be the main cause of DCM, but did have the highest frequency of missense and frameshift variants of all DCM genes. However, it is also true that TTN is the largest gene in the genome and sequencing studies for any disease, even of healthy controls show a high number of missense and frameshift variants. Therefore, although TTN is associated with increased risk of DCM it is not clear that TTN  is the "main" cause of DCM. Please address.

Lines 158-16.

1.  Need to address the point that anthracycline plus trastuzumab risk of HF is significantly greater than anthracycline alone.  Docetaxel plus trastuzumab has better cardiac safety. Also discuss  cardiac safety of TDM1. 

Line 221- chemotherapy and cardiac associated genes: There are now 3 published GWAS studies of chemotherapy-related HF or decline in LVEF: Serie et al 2017, Schneider et al 2017, Wells et al 2017. Discuss that no known CMY genes were in the top hits for any of these studies, but data suggest that rare and common variants in known CMY genes could have modifying effects , Serie et al, J Cardiovasc Dev Dis. 2017 May 4;4(2).

Author Response

We do appreciate the reviewer’s comments on our review of the topic of Genetic factors involved in cardiomyopathies and cancer. Here we enclose point by point answer to the reviewer’s comments.

Response to comments of reviewer 1:

We thank reviewer 1 for the constructive comments.

Comments to the Author

I thank the authors for making the effort to review this interesting and valid topic, particularly to give background on cardiomyopathy genetics and cancer genetics/pathways in the same review. I would suggest the following changes:

Lines 66-81, section on DCM. Points to be addressed:

It is true that DCM was previously (in the past) thought to be secondary to causes other than genetic, but there were several papers suggesting this was not the case even before next generation sequencing technologies were available.

TTN was not shown to be the main cause of DCM, but did have the highest frequency of missense and frameshift variants of all DCM genes. However, it is also true that TTN is the largest gene in the genome and sequencing studies for any disease, even of healthy controls show a high number of missense and frameshift variants. Therefore, although TTN is associated with increased risk of DCM it is not clear that TTN is the "main" cause of DCM. Please address. [1]

[1] We do appreciate reviewer´s comments. Very interesting point. We agree with the reviewer that TTN non-sense variants can be identified in control population. From early studies this finding caused significant debate. Subsequent studies based on analysis of alternative splicing and the domains of the protein, led to a better understanding of the real role of TTN variants in DCM. TTN non-sense mutations in the second half of the gene (approx >150 exons) are usually accepted as disease causing, while the first half of the gene harbors most of the non disease causing variants found in healthy controls.

We have added a new refrence regardind the role of non-sense variants in TTN an patogenicity of DCM. New reference 20 (Roberts AM, et al. Sci Transl Med. 2015. PMID: 25589632)

We have modified the text following reviewer recommendation. “Nevertheless, the genetic factor was supposed to be an important cause of DCM with a genetic test yields around 40% with the current next-generation sequencing panels [18].

The development of high-throughput technologies for genetic testing has led to the identification of new genes associated with DCM as the recognition of titin (TTN) as one of the main DCM associated gene in 2012 [19].”

Lines 158-16.

  1. Need to address the point that anthracycline plus trastuzumab risk of HF is significantly greater than anthracycline alone. Docetaxel plus trastuzumab has better cardiac safety. Also discuss cardiac safety of TDM1. [2]

[2]. We appreciate reviewer´s comment. They are very appropriate. We have included a line on the risk of combination of cardiotoxic agents and the need of new strategies prevention for CCM.

“Discussion on chemotherapy strategies should be taken very cautiously particularly when a combination of potentially cardiotoxic drugs are needed. Future research is warranted for the development of effective preventive medication.”

We have additionally included a new reference [48] De Lorenzo C, Paciello R, Riccio G, Rea D, Barbieri A, Coppola C, Maurea N. Cardiotoxic effects of the novel approved anti-ErbB2 agents and reverse cardioprotective effects of ranolazine. Onco Targets Ther. 2018 Apr 19;11:2241-2250. doi: 10.2147/OTT.S157294.

We believe that further analysis on the cardiotoxicity of possible combination of drugs is beyond the scope of the paper.

Line 221- chemotherapy and cardiac associated genes: There are now 3 published GWAS studies of chemotherapy-related HF or decline in LVEF: Serie et al 2017, Schneider et al 2017, Wells et al 2017. Discuss that no known CMY genes were in the top hits for any of these studies, but data suggest that rare and common variants in known CMY genes could have modifying effects , Serie et al, J Cardiovasc Dev Dis. 2017 May 4;4(2). [3]

[3] Thank you very much for the information on these remarkable papers. We have included the references in the new version of the manuscript and included a new paragraph as suggested (second paragraph in page 7).

“Genome wide association (GWAS) studies have identified a number of loci with enriched variants in patients with CCM compared to controls [85,86]. Candidate markers were not located in known CMY related genes. Further investigation including functional studies is warrant in order to identify implicated genes. Future studies might shed light not only in the pathogenicity of CCM but also on the identification of new CMY associated genes. Interestingly, another recent publication, demonstrated significant association of a number of rare and common polymorphisms in 72 CMY genes in patients with CCM [87]. Within the list of genes, OBSCN seemed to harbor a higher number of missense variants in coding fragments in CCM.“

Reviewer 2 Report

This manuscript provides a comprehensive and up to date review of existing literature about genetic cardiomyopathies including chemotherapy related cardiomyopathy. However, based on the title, the breadth of this paper was not what was expected. Consider a new title that encompasses the scope of the paper.  In addition, the role of genetic counselors and physicians with expertise in genetic cardiomyopathies according to the HFSA/ACMG 2018 guideline should also be noted.

Author Response

Reviewer 2

This manuscript provides a comprehensive and up to date review of existing literature about genetic cardiomyopathies including chemotherapy related cardiomyopathy. However, based on the title, the breadth of this paper was not what was expected. Consider a new title that encompasses the scope of the paper. [1]

 [1] Thank you for your comments, we have changed the title by this one:

“Genetic factors involved in cardiomyopathies and in cancer”

  In addition, the role of genetic counselors and physicians with expertise in genetic cardiomyopathies according to the HFSA/ACMG 2018 guideline should also be noted. [2]

[2] We agree with your comment and we have added this paragraph (lines 412-417):

“The American College of Medical Genetics and Genomics (ACMG) recommend a Detailed 3-generational family history for clinical practice of cardiomyopathy. In addition, referral of patients to expert centers as needed, genetic counseling of patients and families by genetic counselors and physicians with expertise in genetic cardiomyopathies; and therapy based upon phenotype, including drugs, devices, and special clinical recommendations by gene [134].”

New reference [134]. Hershberger RE, Givertz MM, Ho CY, Judge DP, Kantor PF, McBride KL, et al. Genetic evaluation of cardiomyopathy: a clinical practice resource of the American College of Medical Genetics and Genomics (ACMG). Genet Med. 2018 Sep;20(9):899-909. doi: 10.1038/s41436-018-0039-z.

Reviewer 3 Report

The authors provided a comprehensive view of cancer therapy-induced cardiomyopathy (CCM) from multiple aspects, including genetic causes of cardiomyopathies, cancer therapy-related myocardial dysfunction and heart failure, evaluation of cardiac injury from chemotherapy, impact of chemotherapy on systolic impairment development, and predisposition to cancer in patients with cardiac associated genetic variants. They also pointed out the importance of genetic and family study in patients with severe cardiotoxicity. This review is very informative, despite some minor errors.

Minor comments:

  1. Lines 47-48: The tile of the first section may be “1. Genetic causes of cardiomyopathies”, instead of “1. Introduction”.
  2. Lines 71-72: “Genetic cause was supposed to be an important cause of DCM in the childhood, …” could be “Genetic factor was supposed to be…”.
  3. Lines 98-99: “Cardiovascular complications are one of the most frequent of these side effects” should be “Cardiovascular complications are one of the most frequent side effects”.
  4. Lines 221-222: I recommend the title of Section 4 to be changed to “4. Impacts of chemotherapy on development of systolic impairment in patients with germinal mutations at cardiac associated gene loci”.
  5. Line 259: “Elucidate whether an underlying genetic condition is …” should be “Elucidating whether an underlying genetic condition is …”.
  6. Line 283: The subtitle of Section 5, “5.1. Arrhythmogenic cardiomyopathy. Disease of the desmosome” should be “5.1. Arrhythmogenic cardiomyopathy: disease of the desmosome”.
  7. Line 317: There should be only one period at the end of the sentence.
  8. The major content of subsection 5.3 (lines 326-395) is not directly related to cardiomyopathies. So, I recommend shorten or even delete this part.
  9. In Figures 2 and 3, the authors showed some experimental/clinical data. The original literature for these data should be appropriately indicated in the figure legends. If the data were completed by the authors, detailed interpretations may be needed in both main text and figure legends.

Author Response

We do appreciate the reviewer’s comments on our review of the topic of Genetic factors involved in cardiomyopathies and cancer. Here we enclose point by point answer to the reviewer’s comments.

Response to comments of reviewer 3:

Comments to the Author

The authors provided a comprehensive view of cancer therapy-induced cardiomyopathy (CCM) from multiple aspects, including genetic causes of cardiomyopathies, cancer therapy-related myocardial dysfunction and heart failure, evaluation of cardiac injury from chemotherapy, impact of chemotherapy on systolic impairment development, and predisposition to cancer in patients with cardiac associated genetic variants. They also pointed out the importance of genetic and family study in patients with severe cardiotoxicity. This review is very informative, despite some minor errors.

Minor comments:

  1. Lines 47-48: The tile of the first section may be “1. Genetic causes of cardiomyopathies”, instead of “1. Introduction”.
  2. Lines 71-72: “Genetic cause was supposed to be an important cause of DCM in the childhood, …” could be “Genetic factor was supposed to be…”.
  3. Lines 98-99: “Cardiovascular complications are one of the most frequent of these side effects” should be “Cardiovascular complications are one of the most frequent side effects”.
  4. Lines 221-222: I recommend the title of Section 4 to be changed to “4. Impacts of chemotherapy on development of systolic impairment in patients with germinal mutations at cardiac associated gene loci”.
  5. Line 259: “Elucidate whether an underlying genetic condition is …” should be “Elucidating whether an underlying genetic condition is …”.
  6. Line 283: The subtitle of Section 5, “5.1. Arrhythmogenic cardiomyopathy. Disease of the desmosome” should be “5.1. Arrhythmogenic cardiomyopathy: disease of the desmosome”.
  7. Line 317: There should be only one period at the end of the sentence.[1-7]

[1-7] Thank you for all your remarks, we have modified each one from point 1 to 7.

  1. The major content of subsection 5.3 (lines 326-395) is not directly related to cardiomyopathies. So, I recommend shorten or even delete this part.[8]

[8] We consider it important to highlight the association of the genetic basis of cardiomyopathies and cancer. Mutations in desmosomal genes are known to be the genetic cause of cardiomyopathies such as arrhythmogenic and dilated. On the other hand, there are several studies that have suggested the desmosomal genes are involved in cancer progression or prognosis, so we consider the information presented in this section is relevant. However, if the editors consider that we should modify it, we will do it.

  1. In Figures 2 and 3, the authors showed some experimental/clinical data. The original literature for these data should be appropriately indicated in the figure legends. If the data were completed by the authors, detailed interpretations may be needed in both main text and figure legends.[9]

 [9]The figures 2 and 3 were provided by the authors. Both of them are named in the text and the figure legends are explained. We have no included additional information in the text to avoid duplications.
